# Clinical Evaluation of a Multiplex PCR Assay for Simultaneous Detection of 18 Respiratory Pathogens in Patients with Acute Respiratory Infections

**DOI:** 10.3390/pathogens12010021

**Published:** 2022-12-23

**Authors:** Wenmin Li, Xiaoxiao Wang, Wenhao Cui, Leyong Yuan, Xuejiao Hu

**Affiliations:** 1Division of Laboratory Medicine, Guangdong Provincial People’s Hospital, Guangdong Academy of Medical Sciences, Guangzhou 510080, China; 2Department of Clinical Laboratory, Southern University of Science and Technology Hospital, Shenzhen 518055, China

**Keywords:** clinical evaluation, SARS-CoV-2, multiplex probe amplification technique, respiratory pathogens diagnosis, next-generation sequencing

## Abstract

Reliable diagnostics are necessary to identify influenza infections, and coronavirus disease 2019 (COVID-19) highlights the need to develop highly specific and sensitive viral detection methods to distinguish severe acute respiratory syndrome coronavirus 2 (SARS-CoV-2) and other respiratory pathogens to prevent their further spread. In this prospective study, 1070 clinical respiratory samples were collected from patients with acute respiratory infections from January 2019 to February 2021 to evaluate the diagnostic performance of a multiplex probe amplification (MPA) assay, designed to screen 18 pathogens, mainly those causing acute respiratory infections. Ninety-six positive samples and twenty negative samples for the 18 respiratory pathogens defined by the MPA assay and reverse transcription polymerase chain reaction (RT–PCR) were further confirmed by reference next-generation sequencing (NGS). The sensitivity, specificity, positive predictive value (PPV) and negative predictive value (NPV) of the MPA assay were 95.00%, 93.75%, 98.96% and 75.00%, respectively. Additionally, the co-infection rate for these positive samples were 25% (24/95). The MPA assay demonstrated a highly concordant diagnostic performance with NGS in the diagnosis of 18 respiratory pathogens and might play an important role in clinical respiratory pathogen diagnosis.

## 1. Introduction

Acute respiratory tract infections (ARTIs) are among the leading causes of death worldwide [1]. Identifying ARTIs has become a key strategy for limiting the spread of diseases. Since December 2019, severe acute respiratory syndrome coronavirus 2 (SARS-CoV-2) has infected and reinfected more than 637 million people and 6.6 million deaths have been reported globally [2]. The emergence of this highly contagious virus, which can cause asymptomatic infections, has stimulated the development of many rapid, reliable and easy-to-implement diagnostic methods for the early diagnostic of SARS-CoV-2 infection [3,4,5]. However, because of the large overlap in symptoms between coronavirus disease 2019 (COVID-19) and ARTIs, it is difficult to diagnose respiratory pathogen infection based solely on symptoms [6]. Moreover, multiple potential pathogens frequently co-infect patients with SARS-CoV-2, increasing the difficulty of diagnosis. Hence, it is important to develop multi-pathogen diagnostic methods to improve detection efficiency and help the government monitor the spread of these respiratory pathogens when they are circulating [7]. It was observed that for patients with the same length of stay, the outcome of coinfected patients is more severe than that of single mono-infected patients, and the cost of treatment is also higher. It is of great significance to develop rapid and effective diagnostic tests, which can not only reduce unnecessary testing costs for respiratory pathogens and shorten the length of hospital stay for patients, but also reduce the use of antibiotics and the financial burden of hospitalization [8,9]. However, only one pathogen can be detected in a simplex RT–PCR, which is time-consuming, laborious and costly when multiple respiratory viruses need to be detected. Therefore, a highly specific and sensitive viral detection method is needed to distinguish SARS-CoV-2 and other respiratory pathogens, especially methods for the simultaneous detection of SARS-CoV-2 and other respiratory pathogens that cause similar symptoms. In this study, a novel rapid, sensitive and specific diagnostic tool for multiple respiratory pathogens based on the multiplex probe amplification (MPA) assay was developed for screening SARS-CoV-2 and respiratory pathogens in humans with acute respiratory illness. This method is the first respiratory pathogen diagnosis panel to simultaneously detect SARS-CoV-2 and 17 other common respiratory pathogens using the MPA system, namely, influenza A (IFA), influenza B (IFB), parainfluenza virus 1 (PIV1), parainfluenza virus 2 (PIV2), parainfluenza virus 3 (PIV3), parainfluenza virus 4 (PIV4), human adenovirus (HADV), Mycoplasma pneumoniae (MP), Chlamydia pneumoniae (CP), rhinovirus (HRV), respiratory syncytial virus (RSV), human bocavirus (HBOV), human metapneumovirus (HMPV), coronavirus 229E (COV-229E), coronavirus HKU1 (COV-HKU1), coronavirus NL63 (COV-NL63), and coronavirus OC43 (COV-OC43). In addition, the clinical performance of the MPA assay was evaluated using 1070 respiratory specimens from patients with ARTIs.

## 2. Materials and Methods

### 2.1. Specimen Collection

A total of 1070 patients with suspected respiratory pathogen infections were recruited and respiratory specimens were obtained by trained staff using a uniform protocol performed at Guangdong Provincial People’s Hospital and Guangdong Second Provincial General Hospital between January 2019 and February 2021. Specimens were collected in nucleic acid preservation solution containing viral inactivators within 24 h of hospitalization when possible. The specimens were temporarily stored at 4 °C after collection, and transported on cold packs to the Guangdong Provincial People’s Hospital laboratory. Archived residual respiratory specimens were stored at −80 °C and thawed for reanalysis with the MPA assay, RT–PCR and NGS. Upon arrival at the laboratory, specimens were aliquoted and either tested immediately or stored at −80 °C until testing could be completed. All research procedures were carried out in accordance with national ethics regulations and approved by the Research Ethics Committee of Guangdong Provincial People’s Hospital, Guangdong Academy of Medical Sciences. 

### 2.2. Extraction and Purification of RNA/DNA

Total RNA/DNA was extracted using an Easy Pure Viral DNA/RNA Kit (Trans Gen Biotech, Beijing, China) from 200 μL of clinical specimens as well as two microliters of MS2-based pseudo-virus particles, as the RT-PCR internal control, and eluted in 50 μL of DNase-free and RNase-free water. Extracted RNA/DNA was stored at −80 °C.

### 2.3. Detection of SARS-CoV-2 and 17 Other Respiratory Pathogens Using Real-Time Quantitative PCR

RT–qPCR was performed following the manufacturer’s instructions using an approved RT–qPCR (Daan Gene) for the ABI COVID-19 QuantStudio Dx real-time PCR system (Applied Biosystems, USA) to detect SARS-CoV-2. Primer and probe sets targeting the ORF1ab and N genes of SARS-CoV-2 were described in detail by Hu et al. [6]. For IFA, IFA-H1, IFA-H3, IFB, PIV1, PIV2, PIV3, PIV4, HADV, MP, CP, HRV, RSV, HBOV, HMPV, COV-2293/NL63 and COV-OC43/HKU1, samples were validated by qPCR using commercial diagnostic kits (Res 13 × kit) and the design of fluorescent probes targeting these different respiratory pathogens were described by Zhao et al. [10].

### 2.4. Detection of 18 Respiratory Pathogens (MPA Assay)

Details on fluorescent probes with unique melting properties and the principles of MPA technology have been published by Fu et al. [11]. Following the manufacturer’s protocol, a single MPA reaction was performed using two PCR mixes. In tube 1, working mix 1 contained primers and probes for IFA, IFB, PIV1, PIV2, PIV3, PIV4, HADV, MP, CP, HRV and RSV, and in tube 2, working mix 2 contained primers and probes for HBOV, HMPV, COV-229E, COV-HKU1, COV-NL63, COV-OC43 and ORF1ab and N genes of SARS-CoV-2, and the design of fluorescent probes targeting these different respiratory pathogens were described in Table 1. One microliter of extracted DNA/RNA was mixed with a PCR amplification reaction mix (23 µL) containing buffer (deoxynucleoside triphosphates and Mg2). A total volume of 25 µL in each PCR tube per test was prepared by combining the master mix (Taq polymerase, UNG enzyme, and dUTP) and working mix 1 or working mix 2 (primers and probes). The PCR was run on an ABI 7500 Fast real-time PCR system (Applied Biosystems, Warrington, UK) using the following protocol: stage 1, 55 °C for 10 min, followed by 95 °C for 3 min; stage 2, 46 cycles of 95 °C for 10 s, 60 °C for 45 s and 69 °C for 20 s; stage 3, 95 °C for 10 s, followed by 25 °C for 1 min and 68°C for 15 s; stage 4, 95 °C for 10 s, 25 °C for 1 min, 68 °C for 15 s, and 60 °C for 15 s. Fluorescence measurements in the FAM, HEX/VIC, ROX and CY5 channels were recorded during stage 2 (60 °C for 45 s). The fluorescence emission raw data was continually recorded during the temperature increase procedure (stage 2), and the melting curves of FAM, VIC, ROX and Cy5 channels were generated during the dissociation stage of the PCR reaction (from 25 °C to 68 °C in stage 4). In each PCR, the internal control (CY5 detection channel) and all 18 respiratory pathogens, corresponding to the FAM (IFA, IFB and PIV3 in tube 1, SARS-CoV-2-N and SARS-CoV-2 ORF-1ab in tube 2), VIC (PIV1, PIV2, MP and CP in tube 1, COV-HKU1 and COV-NL43 in tube 2), and ROX channels (RSV, PIV4, HRV, HADV in tube 1, HBOV, HMPV, COV-229E and COV-OC43 in tube 2) were simultaneously evaluated. The fluorescent channel and melting temperature of each respiratory pathogen probe are described in Appendix A. Data was analyzed using the Applied Biosystems^®^ 7500 Fast System SDS software with a single threshold to determine the quantification cycle. Samples were retested when the dye signal value of the ROX, FAM, and VIC fluorescent channels were in the gray area (35 < C_T_ < 37), and the test result was considered invalid if the sample had a C_T_ value higher than 36 for cellular DNA. If any sample had a C_T_ value less than 35 in any of the ROX, FAM, and VIC fluorescent channels, the sample was considered positive for the corresponding respiratory pathogens. External positive and negative controls are included in each run of the MPA assay to assess run validity and recognize the cross-contamination.

### 2.5. Validation of the MPA Assay Using NGS

We next validated the MPA assay in 116 samples (96 samples positive by the MPA assay, and 20 samples highly suspected to be positive but once undetermined and redetermined to be negative by the MPA assay) by reference next-generation sequencing (NGS). Data were initially analyzed based on an in-house pipeline produced by Gens Key Medical Technology. Raw sequences were trimmed based on quality and filtered if they were shorter than 130 bases using fastpv0.19.5. First, the sequence reads were screened against the human reference genome and then Bowtie v2.2.4 was used to compare the sequence with the reference genomes of respiratory pathogens. The mapped reads were assembled with SPA des v3.14.0 following the manufacturer’s recommendations.

### 2.6. Statistical Analysis

The detection sensitivity, specificity, positive predictive value (PPV) and negative predictive value (NPV) of the MPA assay were determined based on the NGS results for 116 respiratory specimens. The diagnostic efficiency of the MPA assay was calculated based on the results for 1070 respiratory specimens compared to those with commercial RT-PCR kits. Statistical analysis was performed using kappa concordance coefficients, and statistical significance was set at 1% (*p* < 0.01)

### 2.7. Ethics Statement

Written informed consent was obtained from all participants prior to the study, and the study was approved by the ethics committee of Guangdong Provincial People’s Hospital, Guangdong Academy of Medical Sciences. Using samples collected during clinical-standard COVID-19 testing, the analysis was carried out with no additional burden on patients.

## 3. Results

Characteristics of the subjects. From January 2019 to February 2021, 390 nasopharyngeal swabs (36.45%) and 680 pharyngeal swabs (63.55%) from 1070 patients with a suspected ARTI were collected for the study. Among those patients, 453 were female (42.34%), and 617 were male (57.66%). The median age was 31.75 years (ranging from 5 days to 101 years). Most of the patients presented primarily with fever and cough /expectoration, and some reported with muscle pain and fatigue. The demographic and initial clinical characteristics of the patients are provided in Table 2.

A total of 1070 respiratory tract samples were collected and detected by both the MPA assay and commercial PCR kits. In total, 116 respiratory samples were further validated by reference next-generation sequencing (NGS), including 96 samples that tested positive for one of 18 respiratory pathogens by the MPA assay, as well as twenty samples with dye signal values in the gray area for the first-round MPA reaction that were retested and found to be negative by the MPA assay (Figure 1). The overall agreement between the MPA assay and NGS was 94.83% (kappa = 0.803), with a sensitivity of 95.00%, specificity of 93.75%, PPV of 98.96% and NPV of 75.00%. Most of the positive samples were positive for SARS-CoV-2 (*n* = 28), MP (*n* = 25), RSV (*n* = 21) or IFA (*n* = 11). For PIV-1, PIV-4, HBOV, COV-NL63 and COV-HKU1, no positive specimens were found (Table 3). There were eight cases showing conflicts between the results of NGS and the MPA assay. One sample tested positive for SARS-CoV-2 by NGS but negative by the MPA assay, while another sample was positive for SARS-CoV-2 by the MPA assay but negative by NGS. Three samples positive for IFA and one positive for CP by NGS were tested negative by the MPA assay. In addition, one sample tested true-positive for MP and RSV but false-positive for PIV-2, and one sample tested true-positive for MP but false-negative for HADV by the MPA assay (Table 4).

According to the MPA assay and NGS results, there were 20 dual-positive samples and 4 triple-positive samples. Most of the dual-positive samples were positive for MP and RSV (13/20). Two samples were dual-positive for SARS-CoV-2 and rhinovirus (HRV) (2/20), and two were HADV+MP (2/20). Two were triple-positive for PIV2+RSV+MP; one was positive for HADV+RSV+MP and another was positive for HADV+HRV+MP (Figure 2).

## 4. Discussion

Rapid and reliable diagnosis plays an important role in the containment of respiratory pathogen infections, especially for COVID-19 outbreaks. In this study, we described a rapid and efficient multiplex real-time assay to rapidly diagnose 18 respiratory pathogen infections, including SARS-CoV-2 infection. We calculated the diagnostic efficiency of the MPA assay for the detection of the 18 respiratory pathogens by using 1070 clinical samples. The robustness study suggested that the MPA assay was useful for the detection of respiratory pathogens and revealed the circulating prevalence of some respiratory pathogens.

It must be admitted that the overall positivity rate was approximately 9% (95/1070), somewhat low for patients with acute respiratory infections when testing out a new assay. Since the emergence of COVID-19, China has adopted unprecedented comprehensive public health measures, including suspension of schools and mandatory facial mask usage, to contain the spread of SARS-CoV-2 [12]. As a result, the positive rate of respiratory diseases has been drastically reduced, leading to a certain deviation in the specificity of MPA technology because of the decreased sample size of respiratory pathogens-positive participants. Meanwhile, to test the validity of the MPA assay, we also compared the results of MPA assay with those of commercial diagnostic kits (RT–qPCR kits for SARS-CoV-2 and Res 13 × kit for 17 respiratory pathogens). The MPA assay showed a general accuracy of 100% (1070/1070) with commercial RT-PCR kits when determining whether the patient was infected with respiratory pathogens. In the current study, the most commonly detected viral pathogens were SARS-CoV-2 (28/1070), and RSV (26/1070), followed by MP (21/1070), but the rates were much lower than those in epidemiological studies of pathogens causing respiratory infection before the emergence of COVID-19 [13,14,15]. This decline in infection rate is believed to be related to the infection control measures mentioned above to reduce the spread of respiratory viruses. 

In this study, coinfections with multiple pathogens were reported, while the most common respiratory pathogen in the coinfection patients was Mycoplasma pneumoniae, especially in infant and adolescent patients. The prevalence of coinfection was variable among COVID-19 patients in different studies. In the current study, the overall coinfection rate of respiratory pathogens was 25% (24/95), and our codetection of SARS-CoV-2 and other viral pathogens was present at as 7.14% (2/28), which was slightly different from previous reports [8,16,17]. These differences may be related to various factors, such as sample size, sample type, timing of sample collection, collection and methodology.

We note that 5 MPA assay negative samples (which were undetermined in the first-round MPA assay reaction) that were confirmed to be positive by NGS. On the one hand, the diagnostic sensitivity and specificity of the MPA assay are lower than those of NGS. As a robust tool to obtain extensive genetic information, NGS is a more sensitive, informative and expensive method for the diagnosis of various infections, with a limit of detection (LOD) values as low as 10 copies/mL for respiratory pathogens, including for challenging samples with a low viral content [18]. In the MPA assay, the LOD value was 1000 copies/mL for these 18 respiratory pathogens. However, NGS was too expensive to be performed for all 1070 specimens, which was also the reason why only 116 samples were tested by NGS. On the other hand, our study has a limitation regarding the discrepancy analysis of the IFA results; three samples positive for IFA by NGS tested negative by the MPA assay. We suspected that these three samples did not contain the specific IFA subtypes included in the MPA assay. IFA types pdmH1N1 (2009) and H3N2 are the most common IFA subtypes in China, and thus, the primer and probe of the MPA assay set targeted the corresponding IFA types pdmH1N1 (2009) and H3N2. As a result, the MPA assay may produce false-negative results for other uncommon IFA types, such as H7N9, H5N1 and H5N6 [19,20,21]. In addition, despite the best available reference laboratory tests, contamination issues can still be seen in nucleic acid testing. This explained why some samples that were initially undetermined diagnosed to be negative by the MPA assay. Therefore, it is of great importance to take precautions to prevent cross-contamination to prevent contaminants, such as changing gloves frequently, disinfecting instruments and equipment frequently with ultraviolet light and using an RNA- eliminating spray solution.

It has already been demonstrated by several studies that the MPA system is a good diagnostic tool with high efficiency and rapid diagnostic performance in pathogen detection [22,23,24]. To date, this is the first study to develop a laboratory-based MPA system for the simultaneous detection of SARS-CoV-2 and the other 17 respiratory pathogens. The MPA system in our study demonstrated a highly consistent performance with NGS (kappa = 0.803), with good sensitivity (95.00%) and specificity (93.75%) of the MPA assay in the analysis of 116 clinical samples.

In addition, this rapid and high-throughput method for the simultaneous detection of multiple respiratory viruses based on the MPA platform was compared with other multiplex PCR methods to explore the possibility and practicability limitations. In terms of pathogen coverage, some commercial kits, such as QIA stat-SARS (QIAGEN Diagnostics GmbH) [25,26], Fire RP2.1(Bio Fire Diagnostics) [27,28], can detect more than 20 respiratory pathogens, and they share a similar pathogen spectrum with the MPA assay. However, these commercial kits are much more expensive for each reaction than the MPA assay. Multiplexing 1070 samples can cost as little as 3 USD/per sample for the detection of 18 respiratory pathogens in the MPA assay, while it takes at least 63 USD/per sample to perform 18 respiratory pathogens NGS test. In addition, the analysis of the results can be interpreted automatically and accurately by Applied Biosystems^®^7500 Fast System SDS software, reducing unnecessary manual interpretation errors.

It must be admitted that there are still some limitations in our study. Firstly, the total number of samples positive for respiratory pathogens was small and there was not enough evidence to prove a definitive conclusion. Second, for some targets there was either no positive results (e.g., PIV 1) or only a couple of positives, and more samples should be collected to further confirm the validity of the MPA assay in detecting these respiratory pathogens. However, because of the low infection rate of these respiratory pathogens, few samples have been collected. Finally, as some samples did not contain enough volume, further comparisons of RT-qPCR of 17 respiratory pathogens were unable to be performed, instead, we used the Res 13×kit to detect the 17 respiratory pathogens and we used the NGS result as our gold standard to evaluate the performance of the MPA assay. In conclusion, our developed MPA assay for the simultaneous detection of 18 common respiratory pathogens can help doctors diagnose respiratory pathogens efficiently and provide medication guidance for diseases, making it a useful tool in combating ARTIs.

## Figures and Tables

**Figure 1 pathogens-12-00021-f001:**
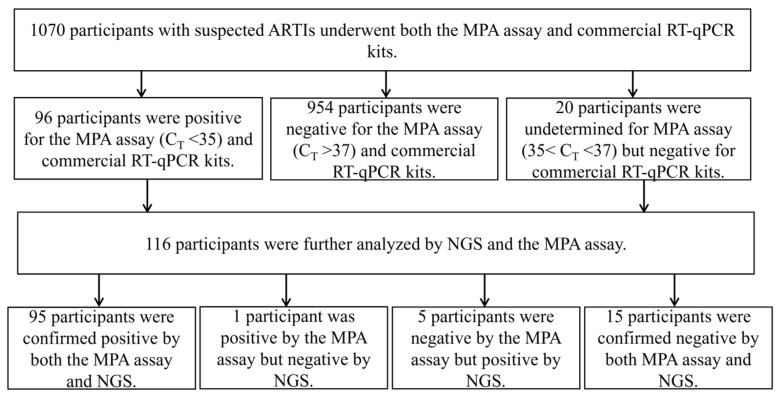
Flowchart of this study: enrollment and outcomes. MPA assay positivity was defined as a RT–PCR cycle threshold (C_T_) value ≤35. Abbreviations: the MPA assay, the multiple probe amplification assay; NGS, next-generation sequencing.

**Figure 2 pathogens-12-00021-f002:**
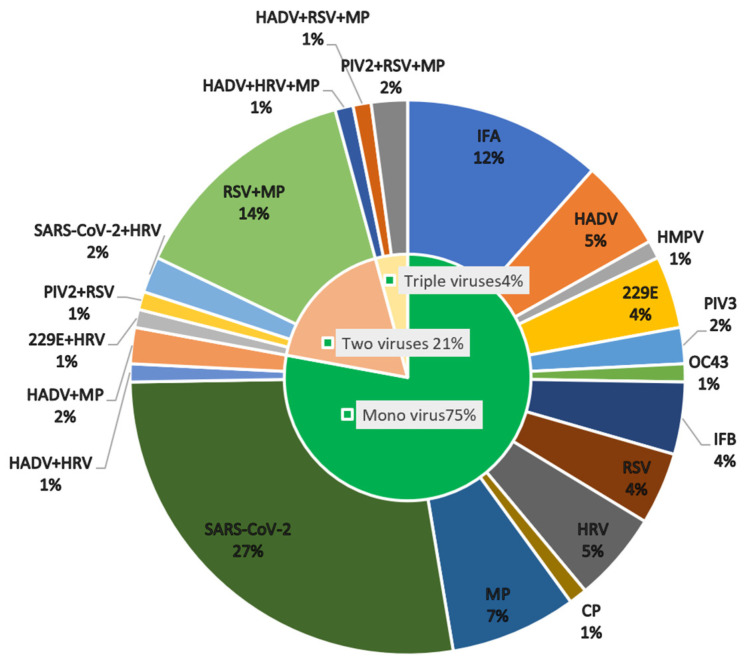
The figure shows the positivity distribution of respiratory pathogens confirmed by both the MPA assay and NGS with a final diagnosis of pulmonary infection. Among the 95 patients, the positivity of patients with one respiratory pathogen infection was 75% (71/95); 21% (20/95) of patients were dual-positive and 4% (4/95) of patients were triple-positive for respiratory pathogens.

**Table 1 pathogens-12-00021-t001:** Information one the MPA Assay primers for sequencing.

Pathogen	Primers	Sequence (5′–3′)	Ampliconsize (bp)
SARS-CoV-2-N	Seq-F	CACCGCTCTCACTCAACATG	120
	Seq-R	CGTCTGGTAGCTCTTCGGTA	
SARS-CoV-2-ORF1ab	Seq-F	GGTGCTTGCATACGTAGACC	120
	Seq-R	ATCACAACCTGGAGCATTGC	
IFA	Seq-F	TCTCATGGAGTGGCTAAAGACA	471
	Seq-R	TGTTCACTCGATCCAGCCAT	
IFB	Seq-F	TGGAGAAGGCAAAGCAGAAC	490
	Seq-R	GACCATCTGCATTTCCCGTC	
RSV	Seq-F	AGGTGGGGCAAATATGGAAAC	440
	Seq-R	TGTCATGTGTTGGGTTGAGTG	
HADV	Seq-F	CATCTCGATCCAGCAGACCT	456
	Seq-R	GATGAGCCGGATCTGACCTG	
HMPV	Seq-F	TGTGCGGCAATTTTCAGACA	549
	Seq-R	TTGTARCAAGCAACCARAGC	
HBOV	Seq-F	AACGTCGTCTAACTGCTCCA	451
	Seq-R	TGCGAGTAGAGTGCCAGTAG	
MP	Seq-F	AAACTGAACCTCCCCGCTTA	499
	Seq-R	TGGCACTACTTGTAGCTGCT	
CP	Seq-F	GATCCTTGCGCTACTTGGTG	600
	Seq-R	GTCTGTTGGCAAGGGGAAAG	
HRV	Seq-F	TGAGGCTAGARATTCCCCAC	242
	Seq-R	AGAGAAACACGGACACCCAA	
COV-229E	Seq-F	GCGTGTTGAAGGTGTTGTCT	520
	Seq-R	TCTGGGGCCAAAACATTGTG	
COV-NL63	Seq-F	GCAGTCGTTCTTCAACTCGT	524
	Seq-R	CTGCTCAATGAACTTAGGAAGGT	
COV-OC43	Seq-F	GCAACAGAACCCCTACCTCT	526
	Seq-R	CGCTGTGGTTTTGGACTCAT	
COV-HKU1	Seq-F	ACTCCCGGTCATTATGCTGG	538
	Seq-R	GAGGCAAAATCGTACCAGGC	
PIV-1	Seq-F	GGCCAAAGATTGTTGTCGAGA	215
	Seq-R	GTTGCAGTCTGGGTTTCCTG	
PIV-2	Seq-F	AGCACGGGGTTCCTATGTYA	544
	Seq-R	TGCTGCTTTGTGATTGGTGT	
PIV-3	Seq-F	CAGAACCCCGTCCTTAGTGA	492
	Seq-R	CACCCAGTTGTGTTGCAGAT	
PIV-4	Seq-F	CAGGCCACATCAATGCAGAA	572
	Seq-R	AAGAACGCACTCATTCCGAC	
RNP	Seq-F	TGCATAACTGTAACAGAGAAACTACCA	148
	Seq-R	TGAGGGCACTGGAAATTGTATAC	

**Table 2 pathogens-12-00021-t002:** Demographics and baseline clinical characteristics of the patients and clinical samples.

Patients	Number (%) (*n* = 1070)
Age Median [IQR]	31.75 (2–58)
Male	617 (57.66)
Signs and symptoms at admission	
Fever	637 (59.53)
Cough	650 (60.74)
Expectoration	441 (41,21)
Fatigue	94 (8.79)
Muscle or body aches	72 (6.73)
Headache	63 (5.89)
Sore throat	74 (6.92)
Chest pain	39 (3.64)
Shortness of breath	245 (22.90)
Runny or stuffy nose	109 (10.19)
Diarrhea	30 (2.80)
Nausea or vomiting	59 (5.51)
Trouble breathing	65 (6.07)

**Table 3 pathogens-12-00021-t003:** Performance of MPA assay for the identification of respiratory pathogens confirmed by sequencing in 116 specimens.

Pathogens	No. of Specimens with Indicated MPA Assay/NGS Result				Performance of the MPA Assay Compared with NGS					
	+/+	+/−	−/+	−/−	Se (%) [95% CI]	Sp (%) [95% CI]	PPV (%) [95% CI]	NPV (%) [95% CI]	Kappa	McNemar Test (*p*)
COVID-19	28	1	1	86	96.55	98.85	96.55	98.85	1	0.954
IF A	11	0	3	102	78.57	100.00	100.00	97.14	0.866	0.250
IF B	4	0	0	112	100.00	100.00	100.00	100.00	1	1
PIV 1	0	0	0	116		100.00		100.00		
PIV 2	3	1	0	112	100.00	99.12	75.00	100.00	0.853	1
PIV 3	2	0	0	114	100.00	100.00	100.00	100.00	1.000	1
PIV 4	0	0	0	116		100.00		100.00		
HADV	9	0	1	106	90.00	100.00	100.00	99.07	0.943	1
MP	25	0	1	90	96.15	100.00	100.00	98.90	0.975	1
CP	1	0	1	114	50.00	100.00	100.00	99.13	0.664	1
HRV	10	0	0	108	100.00	100.00	100.00	100.00	1	1
RSV	21	0	0	95	100.00	100.00	100.00	100.00	1	1
HBOV	0	0	0	116		100.00		100.00		
HMPV	1	0	0	115	100.00	100.00	100.00	100.00	1	1
COV 229E	5	0	0	111	100.00	100.00	100.00	100.00	1	1
COV HKU1	0	0	0	116		100.00		100.00		
COV NL63	0	0	0	116		100.00		100.00		
COV OC43	1	0	0	115	100.00	100.00	100.00	100.00	1	1
**Total**	**95**	**1**	**5**	**15**	**95.00 [88.17–98.14]**	**93.75 [67.71–99.67]**	**98.96 [93.51–99.9]**	**75.00 [50.59–90.41]**	**0.803**	**0.219**

The numbers of positives and negatives detected by both the MPA assay and NGS are shown. (+ = positive), (− = negative), (Se: sensitivity = true positives, TP), (Sp: specificity = true negatives, TN), (PPV = TP/TP + false positives), (NPV = TN/TN + false negatives).

**Table 4 pathogens-12-00021-t004:** Discrepant results between the MPA assay and NGS.

Specimen Number	MPA	NGS
7806	SARS-CoV-2N/ORF1AB (35.3)	−
2877	−	SARS-CoV-2-ORF1AB
2804	−	CP
2840	−	IFA
512	−	IFA
522	−	IFA
2809	PIV2/MP/RSV	MP/RSV
2831	MP	MP/HADV

− = negative.

## Data Availability

Supporting data is being reviewed in the Research Data repository (RDD website: http://183.236.15.75:19800; RDD number: RDDB2022532086).

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
