# Peer review of "Clinical Evaluation of a Multiplex PCR Assay for Simultaneous Detection of 18 Respiratory Pathogens in Patients with Acute Respiratory Infections"

_pathogens, 2022, doi:10.3390/pathogens12010021_

Round 1

Reviewer 1 Report

The article is well written, and the methods and results are described in detail. The article can be improved by further discussing the limitation of the study, the clinical benefit of the differential molecular diagnosis of respiratory pathogens, the relevance of it in the context of SARS-COV 2 pandemic, and the constraints for the implementation of the new method in current clinical practice.

Author Response

We greatly appreciate your professional review work on our paper. In accordance with your useful comments, we have added this point to the Discussion section of the revised paper, please see , lines 312-325 page 10 of the revised paper.

Author Response

  1. Quoting reviewer: In this context, this paper seems valuable. Commercial-based multiple assay system claims for the convenience with less personnel fee, that is “easy to use”, at the expense for the cost. It is better to argue this point in the discussion.”

Response: Thank you for your suggestion. We have added a brief discussion about this point to the Discussion part, please see lines307-309, page 10 of the revised paper.

  1. Quoting reviewer: Line 171; 680 or pharyngeal swabs âž¡ 680 pharyngeal swabs

Figure 1 MPA assay and and commercial RT-qPCR kits âž¡ MPA assay and commercial RT-qPCR kits assay (Ct>37) and and commercial RT-qPCR kits âž¡ assay (Ct>37) and commercial RTqPCR kits

Response: We are sorry for our careless and grateful for your detailed and responsible work on our manuscript. The paper and figure 1 have been improved.

Reviewer 3 Report

General Comments:

The manuscript is in a similar vein to other recent papers looking at the detection of COVID and other resp agents in the context of acute resp infections, using a lab-developed assay.

Overall while it has reasonable readability, there are many areas, some listed below, that need to be addressed together with proof reading by someone with good English skills, as there are numerous spelling and grammatical errors. I also have some concerns about the NGS methodology and its apparent high sensitivity relative to the PCR assays.

Title: self-explanatory

Abstract: adequate

Introduction:

1>   Pg 1, line 40 – PCR testing in asymptomatic persons is no longer a recommendation

2>   Pg 2, line 50 - Ref 8 refers to outcomes in a pediatric population, whereas the patient demographics in this study suggest these are mostly adults, hence this reference is not really applicable ?

3>   Line 61 – why did the authors not include Bordetella pertussis in this panel as this can be another pathogen causing outbreaks in children and adults. Legionella pneumophila can also cause severe acute respiratory infections and is often overlooked.

Methods & Materials:

1>   Were all samples collected into a transport medium ?

2>   Pg 3, line 95 – sub section 2.3 – “detection of COVID + 13 resp pathogens. The authors state this is a commercially available assay that detects these agents. However, lines 98 to 100 list 11 targets, 12 if one includes COVID but not the 13 as in the sub-header?

3>   Pg 3, line 116 – what is a rmalprofile ?

4>   Pg 3, line 138, how did you measure the cellular DNA as there are no listed primers for this target? Also can you clarify the meaning of line 137 to 139 – does the Ct values <36 apply to both the resp target & cellular DNA ? If the Ct value is > 36 did you reextract the sample to verify?

5>   Pg 4 – Statistical analysis lines 160 to 162 – I am unclear in the interpretation of >0.75 which is “deemed good” vs >0.9 which is “considered good” as there is a significant difference between these values and as well the interpretation.

Results:

1>   Pg 4, lines 175 to 177 – What is the significance of patients with tumors and atherosclerosis ? are these patients on immunosuppressive therapies that would predispose them to more severe infections or infections with multiple etiologies, otherwise Table 1 can be shortened.

2>   Fig 1 - the overall positivity rate was about 9% somewhat low for patients with acute respiratory infections when testing out a new assay ?

3>   Pg 6, lines 190 to 193 – overall the positive rate for your samples was low and the manifestations in Table 1 are consistent with a resp infection therefore what key clinical symptoms did you use for the 20 MPA negative samples to decide to retest these by NGS ?

4>   In Table 3, for some targets there are either no positive results (e.g., PIV 1) or just a couple of positives, how valid or useful  are the CIs in these contexts ?

5>   In my experience NGS sequencing requires a reasonable viral load to yield a result, whereas RT-PCR & PCR assays are generally much more sensitive at detecting positives. I would place the MPA in the PCR sensitivity range so how do you explain the positive NGS findings, could contamination have occurred at some point ? Did you retest these samples in the commercial RT-qPCR kits to verify these findings – Tables 3 & 4?

6>   Pg 8 - Figure 2 is very difficult to read because of the colours chosen

7>   Pg 8 - Scheme 3 is different to the information in the text where the text reads that SARS-CoV-2 primers and probes were in tube 2 whereas this Table appears to show they were Tube 1 ?

Discussion:

1>   Many studies would show that during COVID outbreaks, the detection of other respiratory pathogens was severely curtailed or absent, which is the period when these samples would have been collected (Jan 2019 to Feb 2021). Hence the detection of multiple infections (Fig 2), more often seen in children is unusual. How do you explain these findings ?

2>   Pg 9, 247 to 249 – as stated above, NGS in our hands requires a significantly higher viral load than PCR assays to yield a result, and the two references quoted (18 & 19) pertain to non-respiratory pathogens. How do you explain the apparently high sensitivity of your NGS ?

3>   Pg 9, line 252 – you state that you likely encountered 3 samples with non-seasonal subtypes of influenza A? Did you not wish to verify this discrepancy further to determine if it is an assay problem or some other reason?

4>   Pg 9, lines 259 to 263 – you touch on the issue of cross-contamination between samples or contamination in general. In a diagnostic lab, this problem could result in inappropriate patient cohorting and /or treatment. How confident are you in the robustness of the MPA assay that cross-contamination can be recognized?

5>   If this assay is proposed as a diagnostic assay the numbers of positive targets is not sufficient for statistical confidence for any of these parameters apart from possibly the NPV. As well it would also require limit of detection analyses to show the endpoints for these targets, otherwise the MPA is just a comparative pilot.

References:

Many need to be proof read to ensure they have the correct citation information

Round 2

Reviewer 3 Report

General Comments:

See comments in Results

Title: not applicable

Abstract: adequate

Introduction: not applicable

Methods & Materials: not applicable

Results:

1>   Pg 4, line 164, ‘median age was 25 yrs … but in Table 1 the median age is 31.75 yr?

2>   Table 2 – what is Primer NRPP ?

3>   Figure 1 – 2nd row of 3 boxes, right hand box states (35< Ct< 37) should it not be 35> Ct < 37 ? otherwise these would be considered positive ?

4>   Pg 6 line 182 …. RSV (n=21) or IFA (n=11) Table 2…. This Table is a list of primers, sequences etc ?

5>   The revised Figure 2 is no clearer than the original version – if you want to show differences then change the colours for the single, double or triple infection as they all blend into one otherwise turn it into a Table as in its current form it does not illustrate any differences as stated my previous reviewer comments

6>   Was there an amplification step for your NGS assay to improve its sensitivity ?

Discussion: adequate

Author Response

"Please see the attachment
